# The Yin and Yang of Type I IFNs in Cancer Promotion and Immune Activation

**DOI:** 10.3390/biology10090856

**Published:** 2021-09-01

**Authors:** Martina Musella, Claudia Galassi, Nicoletta Manduca, Antonella Sistigu

**Affiliations:** 1Dipartimento di Medicina e Chirurgia Traslazionale, Università Cattolica del Sacro Cuore, 00168 Rome, Italy; claudia.galassi@unicatt.it (C.G.); nicomandu123@gmail.com (N.M.); 2Tumor Immunology and Immunotherapy Unit, IRCCS Regina Elena National Cancer Institute, 00144 Rome, Italy

**Keywords:** Type I Interferons, cancer, immunoediting, anticancer therapy, therapy resistance

## Abstract

**Simple Summary:**

The crucial immune stimulatory functions exerted by Type I Interferons (IFNs) in cancer settings have been not only widely demonstrated during the last fifty years but also recently harnessed for therapy. However, depending on the dose and timing, and the downstream induced signatures, Type I IFNs can and do foster cancer progression and immune evasion. Dysregulations of Type I IFN signaling cascade are more and more frequently found in the tumor microenvironment, representing critical determinants of therapeutic innate and adaptive resistance to several anticancer treatments. Understanding when and through which genetic signatures Type I IFNs control or promote cancer growth is extremely urgent in order to prevent and by-pass the deleterious clinical effects and develop optimized innovative (combinatorial) strategies for an effective cancer management.

**Abstract:**

Type I Interferons (IFNs) are key regulators of natural and therapy-induced host defense against viral infection and cancer. Several years of remarkable progress in the field of oncoimmunology have revealed the dual nature of these cytokines. Hence, Type I IFNs may trigger anti-tumoral responses, while leading immune dysfunction and disease progression. This dichotomy relies on the duration and intensity of the transduced signaling, the nature of the unleashed IFN stimulated genes, and the subset of responding cells. Here, we discuss the role of Type I IFNs in the evolving relationship between the host immune system and cancer, as we offer a view of the therapeutic strategies that exploit and require an intact Type I IFN signaling, and the role of these cytokines in inducing adaptive resistance. A deep understanding of the complex, yet highly regulated, network of Type I IFN triggered molecular pathways will help find a timely and immune“logical” way to exploit these cytokines for anticancer therapy.

## 1. Introduction

In the 1990s, the “danger theory” proposed by Polly Matzinger and colleagues upset the traditional view of immunity as response to solely alien microbes and molecules [1]. Henceforth, it seemed clear that immune responses can be triggered by alarm signals released by the body’s own cells following changes in tissue and organ homeostasis or integrity, as it happens during cancer and viral infection. Intriguingly, antiviral and antitumor immunity share common cell-autonomous responses driven by the emission of danger signals, best known as damage-associated molecular patterns (DAMPs), which actively contribute to the establishment of a productive and long-lasting immune response allowing to clear cancer cells and viral invaders [2]. Among the many DAMPs, cell free nucleic acids emerge as key players in the orchestration of both innate and adaptive immune responses [3,4]. Indeed, during cancer cell immunogenic cell death, cell free RNAs were shown to trigger an acute production of Type I Interferons (IFNs) by dying cancer cells. Type I IFNs then activate an autocrine and paracrine circuit that leads to the production of a plethora of interferon stimulating genes (ISGs), among which the CXC-chemokine ligand 10 (CXCL10) acts as a potent chemoattractant for immune cells. Such Type I IFN signature, so-called “viral mimicry”, on cancer cells was considered a hallmark for the full-blown efficacy of various anticancer treatments [3].

IFNs are a family of pleiotropic immunomodulatory cytokines originally identified as factors capable of interfering with viral replication [5]. Over the years, IFNs were endowed with a plethora of other activities, encompassing a detrimental involvement in autoimmune diseases, metabolic syndromes, and cancer, which make the study of IFN biology crucial for human health and disease. So far, three classes of IFNs (i.e., Type I, Type II, and Type III) were described and distinguished on the basis of upstream *stimuli*, producer cell type, molecular structure, cognate receptor, signaling complex, and molecular function [6]. 

Type I IFNs constitute the largest IFN class [7]. The human genome encodes 17 different Type I IFNs, including 13 subtypes of partially (about 70–80%) homologous IFN-α, and single IFN-β, IFN-ε, IFN-κ, and IFN-ω with lower homology (30–50%) [8,9]. Type I IFN subtypes are produced by all nucleated cells in the body and bind the same heterodimeric cell-surface receptor termed IFN-α/β receptor (IFNAR), formed by the IFNAR1 and IFNAR2 chains, thus triggering the expression of hundreds-to-thousands ISGs [10,11]. Type II IFN family has only one member, IFN-γ, a cytokine that is primarily produced by immune cells, specifically innate-like lymphocyte populations, such as natural killer (NK) cells and innate lymphoid cells (ILCs), and adaptive immune cells, such as T helper 1 (T_H_1) cells and CD8^+^ cytotoxic T lymphocytes (CTLs). IFN-γ signals through the IFN-γ receptor (IFNGR) consisting of the IFNGR1 and IFNGR2 subunits and expressed on most, and almost all, cell types [12]. Type III IFNs are the latest class to be described and include up to four members in humans: IFN-λ1, IFN-λ2, and IFN-λ3, also known as interleukin (IL)-29, IL28-A, and IL-28B, respectively, and IFN-λ4 [13]. Their peculiarity is to be structurally similar to IFN-γ but functionally identical to IFN-α/β. Type III IFNs essentially exert their biological activities on epithelial and immune cells by engaging a receptor complex composed of the IFNLR1 (also known as IL-28AR) and the IL-10R2 chains [14]. 

In this review, we focus on Type I IFNs. We first provide a global description of Type I IFN production and signaling pathways and then we explore their dual and complex role in the evolving tumor-host relationship, while summarizing the use and the function of Type I IFNs in oncology and emphasizing the ability of these cytokines to induce cancer adaptive resistance and immune evasion. 

## 2. Type I IFNs: A Complex Signaling Network

Type I IFNs are considered as the core IFN species in all vertebrates. Evolutionary pressure has so deeply shaped these cytokines that genes encoding for Type I IFNs and their receptors were extensively duplicated, and, especially in mammals, they have gradually lost intronic regions [15]. It is well documented that Type I IFNs constitute the heart of a complex cell signaling system made up of upstream and downstream components and instructed to provide an effective forward line of cell-autonomous defense against foreign pathogens and inner dangers (e.g., cancer cells).

### 2.1. Upstream Triggers of Type I IFNs 

As reported above, Type I IFNs are universally produced in the body. More specifically, IFN-β is released by all nucleated cell types, whereas IFN-α is primarily secreted by a unique subset of immune cells, known as plasmacytoid dendritic cells (pDCs) [16]. Under physiological conditions, small amounts of Type I IFNs are constitutively released. This results in a low tonic IFN signaling responsible for tissue homeostasis and “readiness” to tackle environmental challenges [17,18]. In the face of danger signals, as during viral infection and cancer, Type I IFN production rapidly increases following the stimulation of a wide array of innate immune sensors, known as pattern recognition receptors (PRRs), which recognize and respond to exogenous pathogen-associated molecular patterns (PAMPs) and autologous DAMPs, thus providing a first line of host defense and preserving homeostasis [19,20,21]. Currently identified PRRs reside in different subcellular compartments of immune and non-immune cells, such as the plasma membrane, and the cytosol or endosomal compartments [22], and include Toll-like receptors (TLRs), Retinoic acid-inducible gene I (RIG-I)-like receptors (RLRs), nucleotide-binding oligomerization domain (NOD)-like receptors (NLRs), and different DNA sensors [23]. PRR structures and signaling pathways were recently and extensively reviewed elsewhere [24]. Although viral or autologous nucleic acids (sensed by cytosolic and endosomal PRRs) are the key inducers of Type I IFNs, a minority of other molecules, e.g., bacterial lipopolysaccharide (LPS), viral proteins, lipoproteins, peptidoglycans, and mislocalized or misprocessed endogenous proteins, are also recognized via surface PRRs [22,25,26] and lead to Type I IFN-related innate immune responses [23,24]. 

Briefly, at the cell surface, TLR1, TLR2, TLR4, TLR5, and TLR6 sense bacterial cell wall components, such as lipoproteins, LPS, and flagellin [27,28,29,30]. Within the endosomal compartment, Type I IFN production is mainly evoked via TLR3, TLR7, TLR8, and TLR9 responding to double-stranded RNA (dsRNA), single-stranded RNA (ssRNA), and unmethylated CpG DNA, respectively [31,32,33,34]. Noteworthy, among the cytosolic RNA sensors capable of eliciting Type I IFNs, there are the DExD/H box RNA helicases RIG-I and melanoma differentiation-associated gene (protein) 5 (MDA5, also known as interferon-induced helicase C domain-containing protein 1 (IFIH1)) and the NOD-containing protein 2 (NOD2). RIG-I and MDA5 are RLRs primarily sensing dsRNA or 5’ppp-RNA [35,36,37], whereas NOD2 is a NLR that recognizes ssRNA [38,39]. Immune sensing of cytosolic DNA (being either xenogeneic from intracellular pathogens, exogenous from dying cells, or endogenous resulting from mitochondrial breakdown, DNA damage, and replication stress) is driven by a multitude of cytoplasmic sensors, including the Z-DNA binding protein 1 (ZBP1; also known as DAI) [40] and the nucleotidyltransferase cyclic GMP-AMP (cGAMP) synthase (cGAS) [41,42,43]. Several other proteins described to respond to cytoplasmic DNA encompass the PYHIN family members absent in melanoma 2 (AIM2) [44,45] and IFN-γ–inducible protein 16 (IFI16) [46,47], DExH-box helicase 9 (DHX9) [48], DEAH-box helicase 36 (DHX36) [49], and the DEAD-box helicase 41 (DDX41) [50], as well as the nucleases MRE11 homolog, double strand break repair nuclease (MRE11) [51], and the three prime repair exonuclease 1 (TREX1) [52], all elegantly described in Reference [53].

Overall, once activated, these receptors engage redundant signal transduction cascades that involve adapter proteins and lead to the transcriptional induction of diverse immune genes, including those encoding for Type I IFNs [24]. Specifically, signaling immediately downstream of TLRs, RIG-I, MDA5, DHX9, and DHX36 is largely conveyed through adaptor proteins, such as toll-like receptor adaptor molecule 1 (TICAM1, best known as TRIF), mitochondrial antiviral signaling adaptor (MAVS), or MYD88 innate immune signal transduction adaptor (MYD88) [54,55]. At odds, cGAS, DDX41, IFI16, and, at least in some settings, ZBP1 stimulation activates the stimulator of interferon genes (STING) [24]. The association of PRRs with their adaptors usually converges in the activation of the IκB kinase-ε (IKKε) and TANK-binding kinase 1 (TBK1) complex, which, in turn, phosphorylates and activates key members of the IFN regulatory factor (IRF) family, such as IRF3 and IRF7, and the transcriptional factors nuclear-factor-κB (NF-κB) and activated protein 1 (AP1) [56]. Once recruited, these transcriptional factors translocate to the nucleus, where IRF3, NF-κB, and AP1 trigger a first wave of IFN-β production [57,58,59], commonly observed within 1–4 h after stimulation, while IRF7 regulates a positive feedback loop leading to a secondary burst of IFN-α secretion [60]. It is worth considering that Type I IFN genes go through a tight and context-dependent post-transcriptional regulation via RNA regulatory elements (e.g., microRNAs) [61]. Hence, production of Type I IFN-coding messenger RNAs (mRNAs) may not always guarantee their protein level generation [62]. Interestingly, Type I IFN production can also be contextually induced by cytokines generally produced during inflammation, e.g., tumor-necrosis factor (TNF), macrophage colony stimulating factor (M-CSF), and receptor activator of NF-κB ligand (RANKL) [63,64].

### 2.2. Downstream Effectors of Type I IFNs 

Once released, Type I IFNs act in an autocrine and paracrine manner, and through the IFNAR receptor, give rise to “canonical” and “non-canonical” signaling pathways which directly regulate the transcription of a large set of ISGs [65]. While sharing the same highly evolutionary conserved and ubiquitous receptor, Type I IFN subtypes differ for their binding affinity, which inevitably affects downstream signaling and gene expression profiles [66]. More precisely, IFN-α subtypes show low affinity to both IFNAR1 and IFNAR2 chains (about 1–5 µM and 200 nM affinity, respectively), whereas IFN-β binding is much tighter (100 nM and 0.2 nM affinity, respectively) [67,68].

Upon Type I IFN binding, IFNAR1 and IFNAR2 get into close proximity and dimerize. This causes the reciprocal trans-phosphorylation and subsequent activation of the receptor-associated proteins tyrosine kinase 2 (TIK2) and Janus Kinase 1 (JAK1) [69,70]. In the “canonical” Type I IFN signaling pathway, this results in the generation of docking sites for the recruitment of cytosolic SH2 domain-containing proteins, particularly the signal transducer and activator of transcription 1 (STAT1) and 2 (STAT2) [71,72], leading to their tyrosine phosphorylation and dimerization [73,74]. Phosphorylated STAT1 and STAT2 form the canonical heterodimer (pSTAT1-pSTAT2) that associates to IRF9 to complete the heterotrimeric transcriptional complex, IFN-stimulated gene factor 3 (ISGF3) [10]. Activated ISGF3 then moves to the nucleus where it binds to IFN-stimulated response elements (ISREs) within the promoter regions of ISGs. Thus, ISGs exert the immune-regulatory and antiviral actions of Type I IFNs [11,75]. As a clear proof of JAK and STAT relevance in Type I IFN signaling, mutations or loss of function of these components can predispose the host to infection, autoimmune reactions, cancer, and alterations in therapeutic responses [57]. However, accumulating evidence highlights a far more complex and heterogeneous process of activation and regulation of ISGs, thus assessing the existence of parallel “non-canonical” signaling cascades [76]. Indeed, depending on the cell type, Type I IFNs can also activate distinct STAT isomers (either phosphorylated or not) or even non-STAT pathways [65]. In this regard, following continuous exposure to low levels of IFN-β, unphosphorylated ISGF3 (U-ISGF3) complex prolonged Type I IFN responses through high intranuclear levels of U-STAT1, U-STAT2, and IRF9 [77,78]. Along with this, a body of scientific evidence suggests that transcriptional factors containing IRF9 and either STAT1 or STAT2, but not both, are implicated in controlling ISG expression [79,80]. In particular, a more recent study by Platanitis et al. shows that the transcriptional profile of resting-state murine macrophages is regulated by a preformed STAT2-IRF9 complex, whose generation does not require signaling by IFNAR receptor [81]. Type I IFN stimulation induces a rapid molecular switch from STAT2-IRF9 complex to ISGF3, thus “revving-up” the transcription of most ISGs [81]. Additionally, JAKs can phosphorylate and induce the formation of p-STAT1 and p-STAT3 homodimers, where the former is connected with IFN-γ-mediated signaling by binding gamma activated sequences (GAS) on DNA, and the latter indirectly inhibits pro-inflammatory IFN-mediated responses [82,83,84]. At the same time, Type I IFNs were also reported to signal via STAT2-STAT3 heterodimers, STAT4, a complex formed by STAT5 and CRK likeproto-oncogene adaptor protein (CrkL) and STAT6 [85,86,87,88,89]. To add further layers of complexity, Type I IFNs can directly or indirectly activate other signaling factors while interacting with STAT family members. MAP kinase (MAPK) and phosphoinositide 3-kinases (PI3K)/mammalian target of rapamycin (mTOR) pathways were shown to invoke ISG transcriptional activation [90,91,92]. In this context, Unc-51–like kinase (ULK1) was recently described to affect ISG production and Type I IFN-induced biological activity in a STAT-independent manner [93]. Other “non-canonical” molecules include Schlafen (SLFN) family members and Sirtuins, whose additional effects were examined in various solid and liquid malignancies [94,95,96]. In a recent study on glioblastoma, SLFN5 expression was associated with the negative regulation of STAT1-mediated Type I IFN responses and, thus, with the promotion of a malignant phenotype [97]. Even more, in leukemia and lymphoma cells, SIRT2 was found to induce STAT1 phosphorylation at serine 727 by deacetylating cyclin-dependent kinase 9 (CDK9) in a Type I IFN-dependent manner [96]. 

Owing to the importance of Type I IFN-ISG system during virus-related and unrelated diseases (such as cancer), this signaling cascade is tightly regulated in order to ensure host protection while limiting inflammation and tissue damage and preventing autoimmune responses [98,99]. Therefore, multiple layers of positive and negative feedback mechanisms control the strength and duration of Type I IFN responses. Above all, many components of upstream PRR pathways, including receptors and IRFs, are ISGs [100] Then, negative regulation of cytokine-induced signaling relies on the downregulation of cytokine receptors. Indeed, IFNAR receptors are endocytosed and degraded within minutes of their stimulation and this occurs via ubiquitination of IFNAR1 and exposure of a Tyr-based endocytic motif usually masked by Tyk2 in basal conditions [67,101,102]. Yet, when IFNAR1 is expressed at high levels, its phosphorylation, ubiquitination, and degradation are triggered in an IFN/JAK-independent manner [103]. A further feedback mechanism involves the Type I IFN-mediated induction of the ISG ubiquitin specific peptidase 18 (USP18). This protein endowed with ISG15 (a ubiquitin-like protein)-specific protease activity was shown to not be required for IFNAR desensitization but, rather, to potently interfere with the recruitment of IFNAR1 to the ternary complex by binding to IFNAR2 and STAT2, and, thus, to reduce the responsiveness to Type I IFNs [104,105,106,107,108]. If, on the one hand, lack of USP18 results in a persistent, strong Type I IFN signal, as shown for mouse brain that developed destructive interferonopathy [109], on the other hand, reduced USP18 levels can increase antiviral immunity [110]. Last but not least, Type-I-IFNs are also reported to inhibit their own signaling by transactivating members of the suppressor of cytokine signaling (SOCS) family [111], among which SOCS1 represents a potent inhibitor of Tyk2 [112]. 

All these observations could, at least in part, explain the complex nets regulating ISG expression and ensuring the broad spectrum of Type I IFN responses during infection, cancer, and inflammation.

## 3. Type I IFN-Induced Genetic and Epigenetic Signatures

Depending on their upstream *stimuli* and cellular source, each Type I IFN subtype induces unique and partially overlapping patterns of ISG expression, commonly referred to as “Type I IFN signatures” [11,75]. In their simplest definition, ISGs are all those genes transcriptionally activated during Type I, II, or III IFN responses, and, in humans, they approximately constitute the 10% of the genome. These genes undergo up to 100-fold transcriptional increase depending on the IFN dose, cell lineage, and other endogenous and exogenous variables in cellular signaling [113]. 

Although a good portion of ISGs encodes antiviral proteins (e.g., MX1, MX2, MXA, OAS1, IFI6) [114], a significant one is responsible for the various immune-modulatory activities of Type I IFNs. It is not surprising that the most notorious ISGs include chemokines (e.g., CCL5, CXCL10, CCL3, CCL9,CXCL9, CXCL11) which recruit immune inflammatory cells, apoptosis inducers or modulators (e.g., FAS and its ligand FASL, XIAP-associated factor (XAF1), galectin 9, TNF-related apoptosis-inducing ligand (TRAIL), ISG12, death-activating protein (DAP) kinase, phospholipid scramblase), genes required for major histocompatibility complex (MHC)-I/II based antigen presentation pathways, anti-angiogenic proteins (e.g., STAT1, promyelocytic leukemia protein (PML), guanylate-binding protein 1 and 2 (GBP1/2), ISG20), and cluster of differentiation (CD) molecules (e.g., CD40, CD80) [115,116]. A comprehensive list of ISGs is available at http://interferome.org/interferome/home.jspx. accessed on 1 August 2021). 

As is so often the case, the system is still more complex than it seems. Several ISGs are direct targets of IRF1, IRF3, IRF7, NF-κB, or IL-1 potentially leading to multiple pathways by which a single ISG can be induced [117]. Another crucial aspect that should not be neglected is that some ISGs are basally expressed in addition to being IFN-inducible, while others appear to be expressed only during an IFN response [76]. Yet, although we typically consider ISGs as IFN-inducible protein coding mRNAs, it is important to recognize that IFNs also induce myriad of noncoding RNAs, including long noncoding RNAs and microRNAs (i.e., micro-RNA-106 or miR-106, miR-16) [118,119]. Moreover, a fraction of genes is also repressed during Type I IFN stimulation, and these were denoted as interferon-repressed genes (IRGs or IRepGs) [120,121]. 

Despite the fact that the nature and the underlying molecular mechanisms of a large majority of ISGs are still unknown, Type I IFN genetic signatures have been extensively investigated in the pathogenesis of several autoimmune diseases, including systemic lupus erythematosus, rheumatoid arthritis, systemic sclerosis, and interferonopathies, in general [122,123]. Remarkably, as we are going to discuss in detail in the next sections, these signatures have earned relevant achievements in cancer because of being potentially associated with favorable patient prognosis [124,125] or paradoxically with therapeutic tumor resistance [126,127].

Nonetheless, it has now become clear that Type I IFNs can also induce the so-called “IFN epigenomic signatures” by directly reshuffling chromatin structure and gene expression [128]. The epigenome or ‘epigenomic landscape’ is defined as the whole and dynamic genome pattern of histone and DNA modifications, chromatin conformation, and transcription factor binding that regulates cell-specific gene expression profiles and responsiveness to environmental *stimuli* [129]. Emerging scientific evidence suggests that Type I IFNs induce extensive remodeling of the epigenome by activating new enhancers (termed latent enhancers), disassembling others, and creating chromatin “bookmarks” that *de facto* modulate chromatin accessibility to signal-activated transcription factors and the whole transcriptional machinery to gene regulatory elements [130,131]. IFN epigenomic signatures seem to be mediated by IFN-activated STATs and by de novo–induced transcription factors, such as IRFs, which bind gene-regulatory elements and recruit chromatin-remodeling enzymes [128]. Interestingly, all these changes not only affect ISGs but also occur at regulatory elements of non-ISGs, including canonical targets of the transcription factor NF-κB that encode inflammatory molecules involved in the priming of immune cells, tolerance and the training of innate immune memory [132]. In particular, over the last few years, the phenomenon of *trained immunity* or innate immune memory is attracting increasingly more scientific interest [133]. Accordingly, the engagement of some innate immune pathways, such as Type I IFNs, can induce a downstream global epigenomic reprogramming that, although it does not involve permanent genetic changes (such as mutations and recombinations occurring in adaptive immune responses), sustains changes in gene expression and cell physiology responsible for mounting resistance and a stronger secondary reaction to reinfections [133]. 

Supporting this idea, a comprehensive epigenomic study in human macrophages showed that Type I IFNs potentiated TNF inflammatory function by “priming” chromatin and increasing the amount of trimethylated histone H3 Lys 4 (H3K4me3) at the regulatory elements of genes encoding for inflammatory mediators [132]. The presence of such altered chromatin states and histone bookmarks that are stable over time was found to induce the sequential occupancy of these genomic locations by IRFs and NF-κB and, thus, to enhance their transcription and responsiveness to subsequent environmental challenges [132]. Accordingly, Kamada and colleagues recently and elegantly demonstrated that IFN-β stimulation confers transcriptional memory to fibroblasts, which was attributed to the faster and greater recruitment of p-STAT1 and RNA polymerase II (Pol II) on ISGs and to the acquisition of chromatin marks by H3.3 and H3K36 trimethylation [131]. 

Overall, Type I IFN-induced epigenomic changes can last for days to weeks and, thus, persist beyond the period of Type I IFN expression and upstream JAK–STAT signaling [130,134]. Such persistence confers transcriptional memory and sustains the expression of ISGs. Conversely, for non-ISGs, although Type I IFN-induced epigenomic changes are often transcriptionally silent, the current model posits that such bookmarking alters the way these genes respond to subsequent stimulation [128]. Since a Type I IFN epigenomic signature can increase and sustain immune inflammatory responses, it seems quite obvious that innate immune chromatin reprogramming can have deleterious effects in the pathogenesis of autoimmune and cancerous diseases [133]. In light of this, further investigations into Type I IFN genetic and epigenetic signatures for individual cell types are required to decipher the molecular mechanisms of chromatin modifications and gene expression, in order to provide novel biomarkers of disease pathogenesis and to develop new powerful strategies for autoimmune disorder and cancer therapy.

## 4. Type I IFNs and Cancer: A Troubled Relationship

During the past few decades, the pleiotropic antitumor functions exerted by Type I IFNs became universally acknowledged [135]. Regardless of their source, Type I IFNs play a pivotal role in the dynamic relationship between the host immune system and cancer by directly and indirectly affecting the different aspects of tumor generation, progression, and treatment. In fact, although generally considered as pro-inflammatory cytokines, Type I IFNs are reported to either restrain or promote tumor growth, and this depends on the duration and intensity of the transduced signaling and/or the unleashed ISG profile in the tumor microenvironment (TME) [136], as further discussed below. 

### 4.1. Type I IFNs and Cancer Immunosurveillance

Type I IFN tumor intrinsic role is well documented in many animal models (see Box 1) and is associated with the ability of these cytokines to regulate a plethora of biological processes including cell proliferation, differentiation, survival, and invasion [137]. Specifically, Type I IFNs can affect cancer cell proliferation and exert a cytostatic action both by prolonging or blocking the cell cycle [138,139]. Indeed, Type I IFNs are reported to upregulate the cyclin dependent kinase inhibitors 1A, 1B, and 2B [140] (CDKN1A, CDKN1B, and CDKN2B, which are best known as p21, p27, and p15, respectively), which lead to a delay of the G1-S phase transition [138,140,141,142], and to activate p38 MAPK [143] or CrkL signaling pathways, the latter, in turn, interacting with the tumor suppressor RAP1A, member of RAS oncogene family (RAP1A) [89,144]. Type I IFNs can also regulate apoptosis by modulating both the extrinsic (death receptor-mediated) and the intrinsic (mitochondrial) routes [145,146,147]. Accordingly, some of the best known apoptotic-related genes, such as FAS, FASL, XAF-1, caspase-4, caspase-8, EIF2AK2 (best known as protein kinase R, PKR), galectin-9, and 2’-5’-oligoadenylate synthetase (OAS), particularly the 9–2 isozyme, are ISGs [148]. Furthermore, a recent breakthrough by Frank and colleagues established an interesting link between mitotic cell cycle arrest and IFN-β in the promotion of necroptosis, a programmed form of necrosis, in apoptosis-resistant cancer cells through the phosphorylation of receptor interacting serine/threonine kinase 3 (RIPK3 also known as RIP3) [149]. Concerning the direct antitumoral functions of Type I IFNs, the induction of cell senescence must be mentioned, as well. Indeed, as a result of the strong genomic instability and DNA damage, often ascribed to cancer cells, Type I IFNs are produced and drive the development of oncogene-induced senescence [150]. In this context, the engagement of the cytosolic cGAS-STING signaling pathway seems to be essential for the establishment of such important tumor-suppressive program [151,152]. 

Beyond their cytostatic and cytotoxic activities, Type I IFNs play a fundamental role in coordinating a harmonized immune response against malignant cells. Pioneering studies performed by Brouty-Boye and colleagues demonstrated that the exogenous administration of crude Type I IFN preparations increased the survival of mice affected by lymphocytic leukemia, regardless of the intrinsic sensibility of cancer cells to IFNs [153]. Since then, multiple studies in both human and mice tumor models highlighted the importance of these cytokines in influencing each step of the cancer immunity cycle [135,154]. In the early 1990s, Ferrantini’s group performed studies with different IFNα1 gene transduced cancer cells (Friend leukemia, B16 melanoma, and TS/A mammary carcinoma) in syngeneic immunocompetent mice and observed an important host-dependent tumor rejection complemented by the development of a robust and protective antitumor immunity mediated by CD8^+^ T cells [155,156]. That said, Dunn et al. proved that Type I IFNs intervene in all the three phases of the immunoediting process as review in Reference [157], with a prominent role in the “elimination” stage [158]. By using immunocompetent mice, they specifically demonstrated that endogenous Type I IFNs were required to reject MCA-induced sarcomas and to prevent the outgrowth of primary carcinogen-induced tumors. Moreover, they also observed that several MCA-induced sarcomas from *Ifnar1*^−/−^ mice were rejected in a T cell-dependent manner in wild-type (*Wt*) mice, thus suggesting that tumors arising in the absence of Type I IFN responses are more immunogenic than tumors growing in IFNAR competent hosts [157]. Along similar lines, Type I IFNs exert powerful immunoregulatory effects on multiple cancer cell types (e.g., melanoma, breast, ovarian and colon cancer) by upregulating the expression of surface tumor-associated antigens [159,160,161,162] via upregulation of MHC-I and -II class molecules [163,164], thus increasing the immunogenicity of cancer cells and making them more vulnerable to immune-mediated recognition and destruction. As confirmation of their antineoplastic activity, several important studies describe Type I IFNs as negative regulators of cancer stemness. Cancer stem cells (CSCs, also known as tumor-initiating cells) represent a small immature cellular population within the tumor mass endowed with some unique properties, such as self-renew capacity, multipotency, therapy resistance, and tumorigenic and metastatic potential, that are responsible for tumor progression, recurrence, and poor prognosis [165,166]. A recent study reported that Type I IFNs limit CSC generation and survival, as demonstrated by the evidence that chronic abrogation of endogenous Type I IFN signaling leads to the emergence of more aggressive breast ALDH1^+^ CSCs in HER2/neu transgenic mice [167]. In accordance, Doherty et al. obtained similar results when studying the role of IFN-β on triple-negative breast CSCs [168]. In support of this, already, in 2009, Yuki and colleagues observed how IFN-β negatively affected proliferation, self-renewal capacity, and tumorigenesis of human glioma-initiating cells by inducing their terminal differentiation into oligodendrocytes via STAT3 activation [169]. Another interesting study also underscored the role of Type I IFNs as repressors of glioma stem-like cells (GSCs) by inhibiting their proliferation and self-renewal potential [170]. Yet, by downregulating STAT1, GSCs evade the inhibitory pressure of Type I IFNs in the TME and fuel tumor outgrowth [171]. Moreover, murine lung cancer cells transduced with *Ifnb1* gene showed a significant decrease in their tumorigenic and metastatic capacity, as compared to control cells [170]. Consistent with these findings, IFN-α has also been reported to specifically target the side population of ovarian cancer cells, a subset of cells endowed with stem-like properties [172]. However, as we are going to explain better in the next paragraph, the contribution of Type I IFN signaling on tumor heterogeneity and CSC maintenance/induction is still debated and provides opposite and contradictory results. 

To date, Type I IFNs have gained big attention as crucial factors boosting and bridging innate and adaptive immunity. In this regard, Type I IFNs behave as activators of multiple immune cells, including DC, macrophages, NK, T, and B cells, to promote antitumor immunity. Briefly, DCs, and particularly conventional DCs, are specialized antigen presenting cells (APCs) with the unique ability to process exogenously captured antigen for presentation on MHC-I molecules to CD8^+^ T cells. A substantial number of scientific findings highlights the importance of Type I IFN signaling in the maturation and activation of DCs by enhancing their ability to cross-prime and activate tumor-specific CD8^+^ T cells through the upregulation of the costimulatory molecules MHC-I, MHC-II, CD40, CD80, CD86 [173] and the retention of antigenic particles engulfed from apoptotic cells [174]. Indeed, mice lacking IFNAR1 receptors in DCs are unable to reject highly immunogenic tumor cells, and DCs from these mice display an impaired antigen cross-presentation ability [175]. Upon tumor antigen presentation, CD8^+^ CTLs, with the help of CD4^+^ T cells, acquire the killing capacity for tumor elimination. Many studies have demonstrated the fundamental implication of Type I IFN signaling in the generation, proliferation, differentiation and activity of antigen activated CD8^+^ T cells [176,177]. Type I IFNs do not only promote CTL survival by increasing Bcl-XL expression and IL-2 production [178] but also contribute to their recruitment into the TME through enhanced release of the chemokines CXCL10, CXCL9, and CCL5 [179,180]. Type I IFNs also directly enhance CD8^+^ T cell responsiveness to cognate antigens [181,182] and stimulate the acquisition of CTL effector function through activating the STAT3-Granzyme B (GzmB) pathway, thus ensuring tumor suppression [183]. While promoting CD8^+^ T cell differentiation into GzmB^+^ effector cells, Type I IFNs concurrently seem to limit the expansion of the newly identified TCF1^+^CXCR5^+^ memory and stem-like CD8^+^ T cell compartment, responsible for sustaining the ongoing T cell responses during chronic viral infections and cancer, and preferentially responding to anti-PD-L1 immunotherapy [184,185,186]. Remarkably, since human lymph nodes are protected from Type I IFN signaling, they represent the long-term reservoirs for maintenance of memory TCF1^high^ CD8^+^ T cells [187]. Type I IFNs were also reported to shape CD4^+^ T cell differentiation, expansion and survival and to reinforce the Th1 lineage commitment [188,189]. Notably, Type I IFNs are known to dampen regulatory T cell (Treg) recruitment and activation, thus contrasting the generation of an immunosuppressive TME [190,191,192]. Concerning their innate immunoregulatory functions, Type I IFNs are responsible for the induction of NK cell cytotoxicity. Although mature NK cells did not require Type I IFNs for cancer surveillance [193], the loss of Type I IFN responsiveness negatively impacts on NK cell antitumor immunity and impairs primary and metastatic tumor clearance in multiple models of breast cancer [194,195]. As confirmation of their pleiotropic role in the TME, Type I IFNs also negatively regulate the accumulation and activity of myeloid-derived suppressor cells (MDSCs) [196,197], while stimulating monocyte differentiation into M1-polarized pro-inflammatory macrophages [198], and the production of various pro-inflammatory cytokines (e.g., TNF-α, IL-1, IL-6, IL-8, IL-12, and IL-18) [199]. To establish a more complete framework, Type I IFNs also promote polarization of tumor-associated neutrophils toward an anti-tumor N1 phenotype [200], increase the cytotoxicity of γδ T and NKT cells, as observed in leukemia and melanoma settings [201,202,203], and enhance the antibody-mediated response by promoting the isotype switching in B cells [204,205].

In addition to a direct impact on immune cells, Type I IFNs have extrinsic effects on tumors by regulating their metabolism and the angiogenic process [206]. In light of their ability to impair endothelial cell proliferation and migration, to promote the infiltration T cells able to remodel the tumor vasculature, and to downregulate the expression of the vascular endothelial growth factor (VEGF), Type I IFNs have been longtime considered as inhibitors of angiogenesis [207,208]. Moreover, despite a detailed description is beyond the scope of this review, Type I IFNs drive the metabolic reprogramming of immune cells in the TME, thus impacting on their unique functions [209]. To give some examples, Type I IFNs simultaneously stimulate fatty acid oxidation and glycolysis in pDCs [209,210], while inhibiting cholesterol synthesis in macrophages in a STING-dependent manner [211].

### 4.2. Type I IFNs and Cancer Immunoescape

At odds with the benefits of Type I IFNs in tumor control, a substantial number of scientific studies also describes pro-tumoral properties for these cytokines. Despite being classically depicted as pro-apoptotic agents, IFN-α and IFN-β can induce cell survival and protect cancer cells against apoptotic *stimuli* by activating the NF-κB pathway in a wide variety of cancer types [212,213]. Type I IFNs also upregulate the survival factors MCL1, the apoptosis regulator BCL2 family member (MCL1) and interferon alpha inducible protein 6 (IFI6, best known as G1P3) in myeloma and estrogen-receptor positive breast cancer, respectively, thus promoting poor patient outcome [214,215]. Furthermore, in lung and head and neck cancers, the interferon induced transmembrane protein 1 (IFITM1) was shown to enhance *in vivo* tumor growth and tissue invasion at early stage [216,217]. 

One of the most important emerging hallmarks of cancer is the ability of cancer cells to evade immune responses. As already described above, Type I IFNs are involved in the immunoediting process, and, while intervening in the phase of “elimination”, they also foster the subsequent stages of “equilibrium” and “escape”. During these phases, heterogeneous and genetically unstable tumors, initially held in check by the immunosurveillance, go through a progressive and constant immune pressure culminating with the selection of poor immunogenic cancer cell variants able to evade immune recognition and/or destruction [158]. Coherently, aggressive tumors escaping immune control present a consistent dysregulation and, specifically, downregulation of Type I IFN production and responsiveness [116]. Along with this, Type I IFNs contribute to the generation of an immunosuppressive TME taking part to a process known as immunosubversion, during which cancer cells skew immune cells toward dysfunctional or immunosuppressive phenotypes [218]. In this scenario, Type I IFNs elicit the suppressive or tolerogenic activity of monocyte-derived DCs by promoting the expression of indoleamine 2-3’-dioxygenase (IDO) and the production of IL-10 and other anti-inflammatory mediators, thus ultimately leading to T cell response attenuation [219,220,221]. Type I IFNs also trigger tumor intrinsic IDO production which enhances the immunosuppressive Treg cell activity [222,223] and, as they were involved in the senescence-associated secretory phenotype [224], potentiate the expression of IL-6, which high serum levels are strongly correlated with immunosuppression, accelerated disease progression and poor overall survival in a variety of cancers [225]. Moreover, prolonged Type I IFN signaling could also promote T cell exhaustion and hence immune resistance by stimulating the upregulation of the immune checkpoint ligand PD-L1 (best known as CD274 molecule) [226,227]. Although this evidence supports the clinical development of the combination of Type I IFNs or Type I IFN-inducing therapies with monoclonal antibodies targeting the PD-1/PD-L1 axis [228], a recent study by Jacquelot and colleagues reported that Type I IFN signaling promotes adaptive resistance to anti PD-1/PD-L1 immunotherapy by upregulating nitric oxide synthase 2 (NOS2) [229]. Type I IFNs also regulate the function of immune cells (such as tumor-associated macrophages (TAMs), neutrophils, and MDSCs) that help form the pre-metastatic niche, thus ensuring a suitable environment for tumor development and dissemination [230].

Importantly, Type I IFNs and the ISG IFN-α-inducible protein 27 (IFI27) were associated with the process of epithelial-to-mesenchymal transition (EMT) in ovarian cancer, leading to increased cancer stemness, tumor invasiveness, and therapeutic resistance [231]. Therefore, contrarily to the CSC inhibitory role of Type I IFNs previously described, other studies point to a role of Type I IFN signaling in the generation and/or maintenance of CSCs. First, IFN-α was reported to foster stem-like properties in oral squamous cell carcinoma cells [232] and to affect the invasive potential of pancreatic ductal adenocarcinoma (PDAC) cells by upregulating CSC markers, such as CD24, CD44, and CD133 [233]. Similarly, IFN-β has also been linked to cancer stemness promotion in PDAC by inducing TAM ISG15 ubiquitin like modifier (ISG15) secretion [234], which has also been related to CSC phenotype induction in nasopharyngeal carcinoma [42]. More recently, Qadir et al. found a robust connection between death receptor CD95/FAS, Type I IFN-dependent phosphorylation of STAT1 and stemness in human breast cancer and squamous carcinoma cell lines, which can, in part, explain the correlation between STAT1 activation and therapy resistance observed in these cancer cells [235]. Remarkably, the stimulation of TLR3 and RIG-I induces in somatic cells an innate epigenetic signature which is associated to a process known as “transflammation” [236], involving chromatin remodeling and subsequent nuclear reprogramming, cell plasticity, pluripotency, and even malignant transformation [237,238]. In line with these data, experiments in breast cancer cells put in evidence that activation of NF-κB and β-catenin signaling downstream of TLR3 promoted the enrichment of a subset of cells with a CSC-like phenotype [239]. Similarly, activation of Type I IFN/IRF7 axis is critical for immunological tumor dormancy in breast cancer, an adaptive and protective mechanism that malignant cells adopt to survive stress conditions of the TME and to drive resistance to chemotherapies that preferentially target proliferating cells [240,241].

As previously outlined, a growing body of literature is interested in defining IFN-related signatures responsible for clinical response to or therapeutic resistance of multiple cancer types. In this regard, a panel of 8 ISGs (*STAT1*, *MX1*, *ISG15*, *OAS1*, *IFIT1*, *IFIT3*, *IFI44*, and *USP18*), constituting the “IFN-related DNA damage resistance signature” (IRDS), was associated with cancer cell-intrinsic resistance to various DNA damaging agents (such as anthracyclines, taxanes, cyclophosphamide, methotrexate, and 5-fluorouracil) and radiation therapy, as well as with poor prognosis in a variety of malignancies, including breast cancer and glioblastoma [126,127]. Notably, IFN-β was endowed with the capability to directly upregulate the expression of IRDS genes via U-STAT1 and IRF9 and, thus, to cause resistance to DNA damage and radiotherapy [78].

On the whole, these observations lend further support to the double-edge sword of Type I IFNs in the context of cancer, exerted by controlling tumor growth and promoting tumor escape (as summarized in Figure 1). In this regard, abundant hi-profile literature associated this dichotomy to the duration and intensity of Type I IFN signaling [98,242]. Specifically, acute induction of Type I IFNs, as during immunogenic chemotherapy, induces strong and productive inflammation [3,243], while chronic Type I IFN production is mainly responsible for immunosuppression and therapy resistance [167,229]. Indeed, the length and the strength of Type I IFN signaling are two main factors defining the patterns of ISG expression [24] and epigenetic modulation [128] and, as such, are two major determinants of either beneficial or detrimental “imprinting” on cancer and immune cells, which, thus, tip the balance toward immune control or immune escape, respectively. 

Big issues that need an urgent solution are when and through which ISGs and epigenetic signatures, Type I IFNs, do paradoxically favor tumor progression. This knowledge could help by-pass ineffective and even deleterious clinical outcomes and develop optimized, informed, and hopefully curative chemo-immunotherapies.

Box 1Mouse models to assess Type I IFN role in cancer: principles and applications.Current in vivo approaches to assess the relationship between cancer and immune cells, their co-evolution and the signatures involved in these dynamics, including Type I IFNs, their upstream triggers and downstream effectors, encompass (*i*) transplantable, (*ii*) carcinogen-induced, (*iii*) genetically engineered, and (*iv*) humanized mouse models. In transplantable models, inbred mice, usually from C57Bl/6 or BALB/c strains, are engrafted with histocompatible cancer cell lines either (*i*) heterotopically, generally subcute into the lower flank, which facilitates tumor growth monitoring by simple visual inspection, palpation and the use of a common caliper, or (*ii*) orthotopically, which mimics tissue site-specific pathology, or even (*iii*) systemically (intraperitoneally or intravenously), which allows to study metastatic dissemination [244]. Although displaying many advantages (e.g., synchronous and fast growth of tumors, reproducibility of data, and low costs), transplantable models are undeniably poorly “realistic” as they lack the heterogeneous and multi-step development of cancer cell variants with related effects on the parallel evolution of immune responses as it occurs during “natural” carcinogenesis [244]. Carcinogen-induced models bypass all these limitations. Indeed, these models allow the development of more realistic tumors through the local treatment with ultraviolet (UV) radiation or chemicals (mainly 7,12-dimethylbenz[a]anthracene (DMBA)/12-O-tetradecanoylphorbol 13-acetate (TPA) and 3’-methylcholanthrene (MCA)) [245]. More advanced technologies of genetic engineering let an improved mimicking of spontaneous tumorigenesis through the transgenic expression of oncogenes and/or the inactivation of tumor suppressor genes in germline cells, so-called genetically engineered mouse models (GEMMs), or, in some somatic cells, so-called non-germline GEMM (nGEMM) [246]. These models allow for studying the mechanisms of oncogenic transformation, cancer evolution and therapeutic responses. At the forefront of preclinical research in oncoimmunology, humanized mice have become increasingly refined and used as they closely recapitulate disease pathogenesis and drug testing in humans [247]. On the whole, all these mouse models have provided important insights into the cancer-host relationship also in association with the removal of essential immune-related genes. Their use have been instrumental in defining and supporting the dual role of Type I IFNs and their ISGs in cancer promotion and cancer suppression [248]. Indeed, cancer cells deficient for cardinal elements of Type I IFN production or response could be implanted inmunocompetent or nGEMM histocompatible mice to assess the cancer cell autonomous role of Type I IFNs [3]. Alternatively, transgenic mice (e.g., *Ifnar^−/−^*, *Tlr3^−/−^*, among the others) could be either implanted with *Wt* cancer cells [3] or treated with carcinogens [248] or genetically engineered to develop spontaneous tumors [167] to study Type I IFN signaling on immune and even stromal cells.Therapeutic and prophylactic procedures are the most common experimental settings applied in these models. In the therapeutic setting, tumors (either *Wt* or engineered to lack or overexpress a selected gene) are treated with specific drugs (either as monotherapy or in combination) and local tumor growth, systemic dissemination and mice survival are considered primary endpoints. In the prophylactic setting (mainly used to assess the immunogenic potential of drugs), in vitro killed cancer cells (either as such or loaded on syngeneic DCs) are injected in the subcute of mice, and, after a latency period of 10 to 14 days, their ability to prevent or control a rechallenge with living cancer cells of the same type represents a robust estimate of the elicitation of an adaptive, tumor specific immune response, i.e., *bona fide* ICD [249]. In both settings, mice showing long-term disease eradication could be further challenged to assess the specificity and memory of immune response [249].Currently, literature is rife with descriptions of novel alternative strategies to in vivo preclinical studies, including ex vivo, in sitro, and the most innovative immune-oncology chips [250,251], which are out of our scope and have been extensively reviewed elsewhere [244].

## 5. Type I IFNs and Cancer Therapy: From Response to Resistance

Type I IFNs have emerged as critical determinants of response and even resistance to several anticancer therapies, including conventional chemotherapy, radiation therapy, target therapy and immunotherapy [116,135]. As above comprehensively described, Type I IFNs exert a plethora of immune stimulatory effects [24], which make these cytokines pivotal regulators of immune surveillance. However, depending on the dose and timing, and the downstream induced signatures, Type I IFNs can and do promote tumor progression and immune evasion [24]. Here, we describe the lab and clinical use of Type I IFNs in oncology as we offer a view of their role in affecting the response to diverse therapeutic strategies.

### 5.1. Type I IFNs and Cytotoxic Therapies 

Type I IFNs were shown to critically contribute to the induction and the perception of cancer immunogenic cell death (ICD) upon anthracycline-based chemotherapy [3]. Indeed, anthracyclines, such as viruses, lead to cancer cell-autonomous, TLR3-mediated, Type I IFN production, that triggers autocrine and paracrine IFNAR-dependent circuits, finally resulting in the expression of various ISGs, including CXCL10 (known to recruit innate immune cells). Of note, in several independent cohorts of breast cancer patients, the ISG MX Dynamin-Like GTPase 1 (MX1) was shown to be upregulated following anthracycline-based therapy and to predict clinical response to treatment [3]. On the whole, these observations indicate that “viral mimicry” constitutes a hallmark of successful immunogenic chemotherapy [3,252]. In line with these findings, blocking of IFNAR1 signaling with monoclonal antibodies, was shown to nullify the benefits of target therapy against the human epidermal growth factor receptor 2 (HER2, also known as ERBB2) and the epidermal growth factor receptor (EGFR) [253,254]. Type I IFNs were also shown to be an added value when combined with the ICD inducer cyclophosphamide as they induce the proliferation and activation of CD8α^+^CD11c^+^ DCs in mice bearing transplantable lymphomas [243]. The synergistic effects of Type I IFNs have been extensively reviewed in Reference [255] and are also effective in combination with radiation therapy. Indeed, in a pioneering study, Burnette et al. showed that local ablative radiation therapy triggers the production of IFN-β by myeloid immune cells infiltrating B16F1 melanomas, followed by a strengthening of tumor antigen cross-presentation and treatment response in *Wt* but not *Ifnar1^−/−^* mice [256]. Subsequent studies confirmed the role of radiation therapy-induced Type I IFNs in stimulating cancer immune surveillance [179,257]. In particular, in a mouse model of colorectal carcinoma, the activation of DCs following radiation therapy was shown to rely on cGAS > STING-mediated cytosolic DNA sensing [257]. In this study, the administration of recombinant IFN-β at the tumor site was shown to restore the efficacy of radiation therapy in *cGas^−/−^* and *Sting^−/−^* mice [257].

### 5.2. Type I IFN Monotherapies

Since the first report on their antitumor effects more than 50 years ago [258], Type I IFNs have been the subject of an intensive wave of clinical investigation, which leads to the approved used, by several regulatory agencies, of natural, unmodified recombinant, and pegylated IFN-α variants (Table 1), these last having longer half-life and persistent bioavailability [57,259]. However, over time Type I IFNs have been replaced by new, more efficient therapies. It is right and proper to note that Type I IFNs were used in cancer medicine when their main mechanisms of action were unknown and were conceived, at high doses, as conventional cytostatic drugs. The relevant off-target toxicity of high-dose Type I IFNs finally dictated the failure of these cytokines as clinical drugs. More recent discoveries of the immune modulatory and antiangiogenic effects of Type I IFNs, opened up a plethora of novel clinical indications, thus repositioning Type I IFN cytokines in new-generation cancer therapy, and particularly immunotherapy [135]. Systemic administration of pegylated IFN-α2b in patients with melanoma has shown beneficial effects attributable to the induction of immune infiltration within tumor lesions [260,261]. A renaissance of Type I IFNs also pertains the treatment of chronic myeloid leukemias. Indeed, the use of pegylated IFN-α together with the tyrosine kinase inhibitor Imatinib, resulted in higher rates of clinical responses and in positive immune modulation [262,263,264,265]. 

More than 450 clinical trials are globally assessing the efficacy and the safety of Type I IFN formulations either as monotherapy or as adjunctive to conventional and new generation strategies (https://clinicaltrials.gov) (accessed date on 1 August 2021). Here, below, we offer a view of Type I IFNs in combination (immuno)therapies.

### 5.3. Type I IFNs and Immunotherapies 

Due to their key role in cancer immune surveillance, Type I IFNs are also considered essential for the effectiveness of various immunotherapies and have entered clinical testing in association with immune checkpoint inhibitors (ICIs), adoptive cell therapies (ACTs), cancer vaccines, and oncolytic virotherapies (OVs). 

ICIs are considered pillars of current anticancer therapies [266]. Despite impressive clinical results, therapy resistance and disease relapse can be observed and mainly rely on cancer cell intrinsic and extrinsic factors within the TME [267]. Defects in Type I IFN signaling were shown to affect the efficacy of anti-PD1 based therapies [228,268,269,270]. Similarly, in mice and patients with triple negative breast cancer (TNBC), age-related immune dysfunction impairing ICI therapeutic benefits was shown to rely on decreased Type I IFN signaling [271]. Accordingly, treatment with the STING agonist 5,6-dimethylxanthenone-4-acetic acid restored the effectiveness of ICI-based therapy [271]. Consistent with these observations, a clinical trial (NCT03010176) testing the combination of the anti-PD-1 antibody Pembrolizumab with the STING agonist MK-1454, in patients with TNBC, yielded better response rates and survival. Accordingly, improved clinical responses were observed in patients with stage II and stage III melanomas treated with the anti-CTLA4 Tremelimumab combined with high-dose IFN-α2b [272].

A growing amount of evidence has given impetus to ACT as a potent antitumor tool [273,274]. T cells are unique in the cancer treatment arena as they directly and specifically kill cancer cells, have long-term persistence, and may be engineered to express high-avidity tumor-specific receptors. Type I IFNs were shown to interpose the processes of graft-versus-leukemia (GVL) and graft-versus-host disease (GVHD) by reducing alloreactive donor T-cell expansion, thus providing protection from CD4-dependent GVHD [275]. Of note, Katlinski et al. recently proposed a Type I IFN-related mechanism of immune escape, which affects the activity of chimeric antigen receptor (CAR) T cell transfer [276]. In more detail, colorectal cancer cells were shown to induce the degradation of IFNAR1 on CTLs, thus impairing their survival. Genetic stabilization of IFNAR1 preserved CTL viability and increased the efficacy of CAR-T targeting the fibrinogen activated protein [276].

Therapeutic cancer vaccines experienced a revival in the last decade. A deeper knowledge of tumor antigens and the development of diverse strategies of antigen delivery, helped the design of novel cancer vaccines to educate immune effector cells about what cancer cells “look like” so that they can be recognized and destroyed, thus reducing and, hopefully, avoiding non-specific or adverse reactions [277]. However, the induction of immune suppression and the acquisition of immune resistance still represent significant challenges. The importance of Type I IFNs in improving the effectiveness of cancer vaccines was provided by various studies. In particular, Fu et al. showed that the use of STING agonists in combination with irradiated GM-CSF–secreting whole-cell vaccine (so-called STINGVAX) enhance the activity of tumor-specific CTLs in diverse murine tumor models [278]. Of note, such tumor infiltrating CTLs in STINGVAX-treated mice showed a significant up-regulation of PD-L1, which suggests the possibility to combine STINGVAX with ICIs [278]. Other clinical studies in melanoma patients receiving peptide-based vaccination (Melan-A/MART-1 and NY-ESO-1) have reported an enhanced DC maturation/activation and CD8 T cell activity when the vaccine was combined with low dose IFN-α [279,280].

Apparently at odds with the traditional view of Type I IFNs as antiviral agents, an ensemble of recent studies have shown that Type I IFNs actively contribute to the induction of antitumor-specific responses also in the context of OV [281]. Indeed, Type I IFN signaling was shown to play a role in inducing inflammatory responses in B16 melanomas locally treated with oncolytic Newcastle disease virus combined with systemic CTLA4 blockade. Of interest, the antitumor effects of OV were appreciable also in distant (non-virally injected) tumors [281]. Similarly, the therapeutic activity of a Semliki Forest virus encoding IL-12 was reported to strongly rely on a vector-induced Type I IFN response in the host [282]. 

### 5.4. Type I IFNs and Therapy Resistance

Although Type I IFNs have improved the clinical outcome in specific therapeutic settings, the development of resistance represents a recurrent problem. The complexity and the magnitude of Type I IFN signaling, together with the plasticity and the heterogeneity of cancer cells and their ever-evolving cross-talk with host immune cells, leads to the insurgence of drug resistance by distinct mechanisms, that we broadly classify as (*i*) unresponsiveness to Type I IFNs and (*ii*) Type I IFN-induced resistance. 

Unresponsiveness to Type I IFNs. A variety of mechanisms were described to affect the sensing of Type I IFN signals. These include: (*i*) degradation of IFNAR1 in cancer and immune cells through ubiquitination driven by the phosphorylation on serine residues. This pathway is triggered by a variety of stress- and inflammation-related signals within the TME (e.g., hypoxia [283], proinflammatory cytokines [284], and VEGF [285]) and leads to tumor progression and immune suppression by altering the expression of the ISGs IFIT2, IRF7, MX2, and USP18 in cancer cells and of IL-2 in immune cells [150,276]; (*ii*) impairment of IFNAR1 signaling through SOCS-mediated prevention of STAT1 phosphorylation [286]; (*iii*) silencing or loss-of-function mutations of JAKs and STATs [157,270,287,288]; and (*iv*) downregulation of IRFs [289,290,291]. 

Type I IFN-induced resistance. Groundbreaking studies by Andy J Minn established the existence of an IFN-related DNA damage resistance signature (IRDS) responsible for an adaptive resistance to chemotherapy and radiation therapy [126]. Subsequent findings confirmed a role for Type I IFNs in adaptive resistance to therapy, as they induce surface PD-L1 on cancer cells [292]. However, prolonged Type I and II IFN exposure, was associated to a state of chronic resistance to anti PD-L1 therapy [227]. Such ICI resistance was described to rely on epigenomic changes on cancer cells, the expression of various ISGs (i.e., *Tnfrsf14*, *Lgals9*, *MhcII*, *Ifit*, *Mx1*), and induction of multiple T cell inhibitory receptor networks [227]. These findings are in line with previous reporting of a persistent Type I IFN related signature in RT-resistant squamous cell carcinomas, and breast, prostate, and glioma cells [293,294]. Of note, Type I IFNs were found to be required for CD95-induced stemness in cancer cells [235]. The mechanisms underlying CSC induction rely on STAT1 phosphorylation and the downstream activation of PLSCR1, USP18, and HERC8 [235]. Along similar lines, exosome transfer from stroma to cancer cells, was shown to drive RIG-I > STAT1-dependent signaling while promoting cancer stemness and therapy resistance [295]. Further confirming that Type I IFNs could stay at the pinnacle of therapy resistance, a more recent study described chemotherapy-induced activation of STING-mediated type I IFN signaling as a cell-intrinsic mechanism of cell survival and regrowth in diverse breast cancer cell lines [296]. On the whole, these works point out the need to dig deeper into viral-like signatures in cancer in order to tailor and optimize therapy. Indeed, therapeutic failure is not an option. Only a deep understanding of the players and the dynamics of cancer adaptation to therapies will help overcame resistance and reach effectiveness as the only possibility.

## 6. Conclusions and Future Perspectives

Type I IFNs are master regulators of cancer immunity as they tip the balance between cancer immune surveillance and cancer immune escape and between therapy response and therapy resistance. The complexity of Type I IFN signaling networks, and the plasticity of their downstream effects within the TME, leaves no doubt that a better understanding of the where, when, and how Type I IFNs should be induced or delivered in tumors in a way that promotes disease control, while preventing deleterious effects, will have invaluable insights in cancer management. Filling this gap of knowledge will indeed help to open the way for the development of innovative (combinatorial) strategies that will yield more effective and durable responses.

## Figures and Tables

**Figure 1 biology-10-00856-f001:**
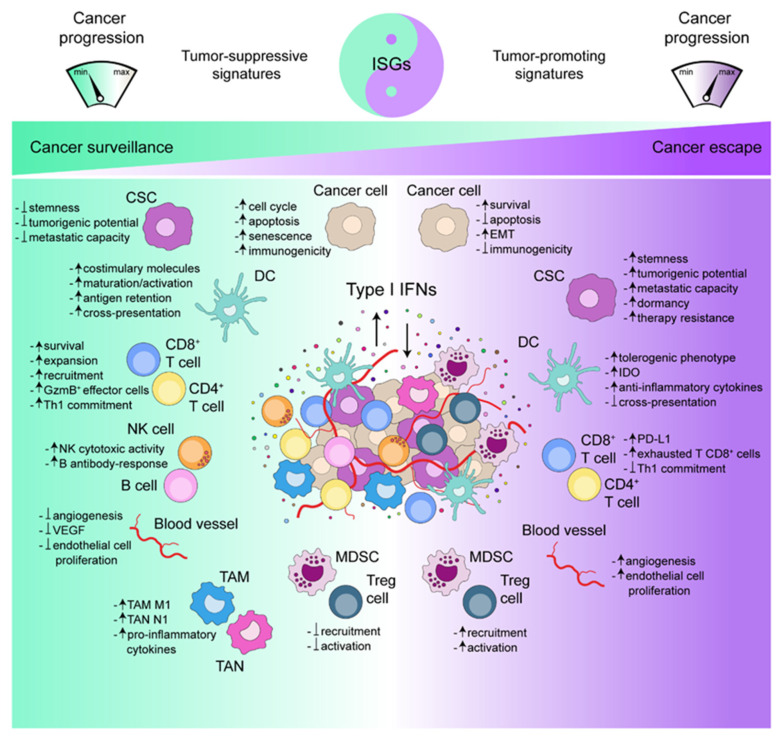
The yin and yang of Type I IFNs in the TME. Type I IFNs can favor cancer surveillance or cancer escape by triggering different ISGs and acting on the various cell types constituting the TME. When the tumor-suppressive genetic signatures prevail, Type I IFNs limit cancer (stem) cell expansion and angiogenesis, while promoting effective antitumor immune responses. Otherwise, the onset of tumor-promoting genetic signatures exacerbates the dark sides of Type I IFNs, thus favoring cancer progression. CSC: cancer stem cell; DC: dendritic cell; GzmB: granzyme B; NK: natural killer; VEGF: vascular endothelial growth factor, TAM: tumor-associated macrophages; TAN: tumor-associated neutrophils; Treg: regulatory T; MDSC: myeloid-derived suppressor cell; EMT: epithelial-to-mesenchymal transition; IDO: indoleamine 2-3’-dioxygenase.

**Table 1 biology-10-00856-t001:** Type I IFN formulations for clinical use.

Name	Company	Type I IFN Subtype	Indication(s)
Alfaferone	Alfa Wassermann	Natural leukocyte interferon alpha	Hairy cell leukemia, multiple myeloma, non-Hodgkin lymphoma, follicular lymphoma, chronic myelogenous leukemia, malignant melanoma, AIDS-related Kaposi’s sarcoma
Belerofon	Nautilus Biotechnology	Interferon alpha	/
HeberFERON	Center for Genetic Engineering and Biotechnology	Interferon alpha-2b/interferon gamma	Basal cell cancer
IFN alfa-2b XL	Flamel Technologies	Interferon alpha-2b	/
Infergen	Amgen	Interferon alfacon-1	Non-Hodgkin’s lymphoma, ovarian cancer
Intron A	Biogen Idec	Interferon alpha-2b	Chronic myeloid leukemia, follicular lymphoma, hairy cell leukemia, malignant melanoma, multiple myeloma, non-Hodgkin’s lymphoma, AIDS-related Kaposi’s sarcoma
Joulferon	Human Genome Sciences	Albinterferon alpha-2b	/
Locteron	Biolex	Interferon alpha-2b	/
Multiferon	Viragen	HuIFN-alpha	Chronic myeloid leukemia, hairy cell leukemia, malignant melanoma
Novaferon	Genova Biotech Company	Interferon alpha	Colorectal cancer, neuroendocrine tumors, pancreatic cancer
Pegasys	Hoffmann-La Roche	Peginterferon alpha-2a	Malignant melanoma, renal cell carcinoma
PegIntron	Enzon Pharmaceuticals	Peginterferon alpha-2b	Malignant melanoma, cholangiocarcinoma, chronic myeloid leukemia, solid tumors
Reiferon Retard	Rhein Minapharm Biogenetics	Peginterferon alpha-2a	/
Roferon A	Hoffmann-La Roche	Interferon alpha-2a	Chronic myeloid leukemia, cutaneous T-cell lymphoma, hairy cell leukemia, Kaposi’s sarcoma, malignant melanoma, non-Hodgkin’s lymphoma, renal cell carcinoma
Sylatron	Merck	Peginterferon alpha-2b	Melanoma
Wellferon	Glaxo Wellcome SA	Interferon alpha-n1	Chronic myeloid leukemia, Hairy cell leukemia

## Data Availability

Not applicable.

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
