# Peer review of "The Yin and Yang of Type I IFNs in Cancer Promotion and Immune Activation"

_biology, 2021, doi:10.3390/biology10090856_

Round 1
Reviewer 1 Report
1) There is a confusion in the abstract regarding the term "DAMP". I think Type I IFNs cannot be qualified as DAMPS as it is stated (line 42-43 "Among the many DAMPs,
Type I Interferons (IFNs) emerge as key players...". For a comprehensive review on DAMPS and their classification, I think some papers authored by WG Land need to be cited (e.g Damage-associated molecular patterns in trauma. Eur J Trauma Emerg Surg. 2020 Aug;46(4):751-775.)
2) line 49 not three main classes of IFNs..but three classes
3) in most exemples that are cited, it is not clear if they are resulting from observations in humans or experimental work in mice. In general, I think that a chapter dedicated to cancer cells implantation in control and IFNAR (or STAT) KO mice should be developed to assess the experimental reality of the roles of type I IFNs in various cancers.
4) table 1 is incomplete, as the impact of anti IFN Ab, Anti IFNAR Ab or even kinoids needs to be described in the frame of cancer....This would more relevant to the present review than the first chapters describing IFN signaling that has been reviewed many times and could be summarized here with a simple figure.
Reviewer 2 Report
The manuscript called: “The yin and yang of Type I IFNs in cancer promotion and immune activation“ is a very comprehensive and in depth review of the ambivalent role of type I interferons in cancer. The review is well written and easy to follow, with all the important references included. I only have minor comments:
On the one hand, immunogenicity of ICD driven by anthracyclins and radiotherapy has been shown to be strongly associated with type I IFN signaling (ref. 3 and ref. 255), on the other hand IFN-related signatures were shown to be responsible for therapeutic resistance (line 517). This discrepancy could be discussed more in detail.
The sequence of chapter 5 would be probably more logical, if chapter 5.2 would be the first one, as this is the only part in chapter 5, which does not include exogenous IFN as a therapeutic agent and describes the association between endogenous IFN/IFN receptors and response to chemo and radiotherapy. The other chapters describe IFN as an therapeutic agent in monotherapy and combination therapy.
There is not clearly stated in the manuscript, what is the driver of the decision, whether type I IFN signaling would be pro- or anti-tumorigenic. I liked the statement in Fucikova et al. (Cell Death and Disease, 2020) that: “Acute, robust type I IFN responses have been associated with immunostimulation, whereas chronic, indolent type I IFN signaling mediates immunosuppressive effects.” Do you consider this statement to be accurate? Could you encapsulate the ambivalent role of type I IFNs to similar conclusion, or is it too simplified?
